

**Effectiveness of Emission Controls on Atmospheric Oxidation**
**and Air Pollutant Concentrations: Uncertainties due to Chemical**
**Mechanisms and Inventories**
Mingjie Kang[1,2], Hongliang Zhang[3], Qi Ying[4,a]
[1] School of Applied Meteorology, Nanjing University of Information Science and
Technology. Nanjing 210044, China.
[2] Atmospheric Environment Center, Joint Laboratory for International Cooperation on
Climate and Environmental Change, Ministry of Education, Nanjing University of
Information Science and Technology. Nanjing 210044, China.
[3] Department of Environmental Science and Engineering, Fudan University, Shanghai
200433, China.
[4] Zachry Department of Civil and Environmental Engineering, Texas A&M University,
College Station, Texas 77843-3136, USA.
[a] Currently at the Division of Environment and Sustainability, Hong Kong University of
Science and Technology, Clear Water Bay, Kowloon, Hong Kong, China.
*Correspondence to*: Qi Ying (qying@ust.edu)





**Abstract.** In this study, three photochemical mechanisms of varying complexity from the
Statewide Air Pollution Research Center (SAPRC) family and two widely used anthropogenic
emission inventories are employed to quantify the discrepancies in the predicted effectiveness
of nitrogen oxides ($NO_x$) and volatile organic compound (VOC) emission controls on ozone
($O_3$), secondary inorganic aerosols (SIA), and hydroxyl (OH) and nitrate ($NO_3$) radicals using
the Community Multiscale Air Quality (CMAQ) model. For maximum daily average 8-hour
$O_3$ ($O_3$-8h), relative reductions predicted using different emission inventory and mechanism
combinations are consistent for up to 80% $NO_x$ or VOC reductions, with maximum differences
of approximately 15%. For secondary inorganic aerosols (SIA), while the predicted relative
changes in their daily average concentrations due to $NO_x$ reductions are quite similar, very
large differences of up to 30% occur for VOC reductions. Sometimes even the direction of
change (i.e., increase or decrease) is different. For the oxidants OH and $NO_3$ radicals, the
uncertainties in the relative changes due to emission changes are even larger among different
inventory-mechanism combinations, sometimes by as much as 200%. Our results suggest that
while the $O_3$-8h responses to emission changes are not sensitive to the choice of chemical
mechanism and emission inventories, using a single model and mechanism to evaluate the
effectiveness of emission controls on SIA and atmospheric oxidation capacity may have large
errors. For these species, the evaluation of the control strategies may require an ensemble
approach with multiple inventories and mechanisms.





## 1. Introduction

Tropospheric ozone ($O_3$) pollution remains a major global concern. $O_3$ plays a critical role in atmospheric chemistry and is an important target for air quality improvement (Lu et al., 2018; Lyu et al., 2023; Real et al., 2024) because high levels of surface $O_3$ negatively affect human health, agricultural crop yields and plant growth (Du et al., 2024; Feng et al., 2022; Ghude et al., 2016; Lu et al., 2020; West et al., 2006). Surface $O_3$ is mainly formed by the photochemical reactions of $NO_x$ and volatile organic compounds (VOCs) emitted from anthropogenic and biogenic sources (Finlayson-Pitts and Pitts Jr, 1999; Seinfeld and Pandis, 2016). The development of effective $O_3$ control strategies is hampered by the considerable spatial and temporal variability of surface $O_3$ concentrations and their non-linear relationship with emissions and meteorological conditions.

Three-dimensional chemical transport models (CTMs) are a valuable tool for developing effective air pollution control strategies. They can provide spatial and temporal information on $O_3$, particulate matter, and toxic air pollutants by numerically solving the mathematical equations describing the emission, reaction, transport, and deposition of primary and secondary atmospheric pollutants (Byun and Schere, 2006; Russell, 1997). The veracity of modeling outcomes is contingent upon the gas-phase chemical mechanisms and emission inventories, among other variables. Reliance on a single chemical mechanism and emission inventory may result in substantial uncertainty in modelled pollutant concentrations. A common approach to reduce uncertainty in air quality model predictions is to use an ensemble of simulations with different emission inventories and chemical mechanisms (Hu et al., 2017a).

Photochemical mechanism is one of the core components of all CTMs. While the representation of inorganic chemistry is generally similar across mechanisms, the representation of atmospheric organic chemistry differs significantly in terms of the number of explicit model species, the lumping schemes, and the radical chemistry, leading to variations in the model predictions of $O_3$, $PM_{2.5}$, air toxics, and some important radical species (Griffith et al., 2016; Kim et al., 2009). Furthermore, the responses of the predictions to changes in emissions may also differ depending on the photochemical mechanism employed, which may impact the assessment of emission control strategies. In practical applications, it is of the utmost importance to strike a balance between mechanism complexity and computational efficiency. For long-term modeling studies of criteria pollutants and the evaluation of numerous emission control strategies, condensed mechanisms may prove to be the optimal choice. More detailed mechanisms are appropriate for a broader range of applications, such as





investigations into specific reaction products that are not explicitly represented in condensed
mechanisms. However, they are more demanding on computational resources.

In this context, several mechanism comparison studies have been conducted. Differences

in predicted $O_3$ levels using various photochemical mechanisms have been reported (Yu et al.,
2010; Venecek et al., 2018). In addition to directly comparing the model predictions of $O_3$
concentrations, comparative analyses were also undertaken to examine the similarities and
differences between these mechanisms in predicting $O_3$ changes in response to changes in
precursor emissions. For example, Li et al. (2012) and Kang et al. (2022) compared several
mechanisms from the SAPRC family, including the standard versions of SAPRC-99, SAPRC-
07, SAPRC-11, and a highly condensed version of SAPRC-07. Their results showed that,
despite discrepancies in the predictions for $O_3$, key radicals such as OH and $HO_2$, and oxidation
products such as $HNO_3$, $H_2O_2$, $NO_2$, PAN, and HCHO, the relative changes in $O_3$ due to
changes in $NO_x$ and VOC emissions were almost identical.

On the other hand, the accuracy of model predictions is also significantly affected by

uncertainties in anthropogenic emission inventories (Hu et al., 2017a; Kang et al., 2022; Placet
et al. 2000), which primarily arise from uncertainties and variability in activity levels (e.g.,
industrial production or energy consumption) and emission factors (Akimoto et al., 2006; Lei
et al., 2011; Streets et al., 2003). For example, when local speciation profiles are not available
for the generation of mechanism-specific VOC emissions from the emission rate of non-
methane hydrocarbons, average speciation profiles from the SPECIATE database developed
by the US EPA were often adopted (Bray et al., 2019; Li et al., 2014; Streets et al., 2003; Wu
and Xie, 2017). However, the emission factors from the SPECIATE profiles are predominantly
representative of the characteristics of local emissions, as a consequence of the disparities in
emission standards and control technologies among diverse geographical regions (Sha et al.,
2021). This introduces uncertainties into the emissions and the predicted pollutant
concentrations.

To date, several emission inventories covering China have been developed, such as the

Multi-resolution Emission Inventory for China (MEIC), the Regional Emission inventory in
ASia (REAS), and the Emission Database for Global Atmospheric Research (EDGAR). These
inventories have been successfully applied in chemical transport modeling to investigate the
concentration and spatial distribution of $O_3$ and other pollutants (Hu et al., 2017b; Kang et al.,
2021; Li et al., 2019, 2018; Saikawa et al., 2017; Xue et al., 2020; Yamaji et al., 2008). Hu et
al. (2017a) reported inconsistencies in emission inventories in predicting $O_3$ and $PM_{2.5}$ using
the Weather Research and Forecasting/Community Multiscale Air Quality (WRF/CMAQ)





model system. Ma et al. (2004) identified variations in $NO_x$ and VOC emissions among
different inventories as the main factors influencing modeled $O_3$ concentrations. Our previous
study, which employed the condensed SAPRC-07 mechanism and two anthropogenic emission
inventories (MEIC and REAS), also observed relatively larger differences in $O_3$ predictions
between inventories in megacities of Beijing and Shanghai, especially on days with elevated
$O_3$ levels (Kang et al., 2022).

While several studies have explored the differences in model predictions due to varying

chemical mechanisms or emission inventories, a comprehensive analysis of the influence of
diverse combinations of these two factors on the sensitivity of $O_3$ and other air pollutants to
emission changes in China has yet to be undertaken. In this study, we address this gap by
applying the CMAQ model, integrated with three photochemical mechanisms and two widely
used emission inventories, to quantify the effects of different combinations of mechanisms and
inventories on the predictions of maximum daily average 8-hour ozone ($O_3$-8h) and other
secondary pollutants in different regions of China. In addition, the impacts on atmospheric
oxidation capacity and key gaseous pollutants are investigated. To gain insight into the
variations in pollutant sensitivity, a series of incremental emission reduction scenarios were
used, thereby enabling the quantification of the influence of different mechanism and inventory
combinations on the response of $O_3$ and related pollutants. The findings of this study can assist
policymakers in the development of more effective and adaptive pollution control strategies.

**2. Materials and methods**
**2.1 The CS07, SAPRC-11 and SAPRC-18 mechanisms**
Three different chemical mechanisms from the SAPRC mechanism family were used in this
study, i.e., the condensed SAPRC-07 (CS07) (Carter, 2010), the standard SAPRC-11 (S11)
(Carter and Heo, 2013), and the standard SAPRC-18 (S18) (Carter, 2020). These mechanisms
were selected to represent different levels of detail in gas-phase reactions in a regional chemical
transport model. The CS07 was derived from the widely used SAPRC-07 mechanism and has
a high condensation level similar to that of the Carbon Bond mechanism, which is also widely
used. The S11 is an updated version of the SAPRC-07 mechanism, with significant revisions
made to the aromatic chemistry. The S11 mechanism employed in this study is identical to that
utilized in our previous study (Kang et al., 2021). For a detailed description of CS07 and S11
regarding $O_3$ source apportionment and emission sensitivity, please refer to Kang et al. (2022).
The S18 mechanism represents a complete update of the SAPRC mechanism since SAPRC-07.
S18 incorporates a greater number of model species, a more explicit representation of peroxyl



radical chemistry, and a lumping scheme that is more suitable for predicting secondary organic
aerosol (SOA) formation. Due to these modifications, S18 is more extensive than S11 in terms
of both the number of species and chemical reactions. Although it has been successfully applied
in photochemical box models (Jiang et al., 2020; Li et al., 2022a, b), it has not yet been
implemented in 3D regional CTMs.

**2.2. Anthropogenic emission inventories**
The present study compares two widely used anthropogenic emission inventories, MEIC
(http://www.meicmodel.org) and REAS 3.1 (https://www.nies.go.jp/REAS/), to investigate $O_3$
pollution in China. The emission data from these inventories were processed using an in-house
emission processor. Detailed VOC speciation profiles selected from the US EPA-developed
SPECIATE database were processed using the speciation profile processor from W.P.L. Carter
(2015) to generate profiles for the CS07, SAPRC-11, and SAPRC-18 mechanisms, which are
used to estimate emissions of CMAQ-ready VOCs. The MEIC emission inventory includes
only emission estimates in China, whereas the REAS emission inventory has complete spatial
coverage for Asian countries. In the MEIC simulation, emissions from other countries are
supplemented using data from the REAS inventory.

**2.3 Model application**
The CS07, S11, and S18 mechanisms were incorporated into the CMAQ model (version 5.0.2)
to evaluate the differences of mechanisms and inventories in predicting $O_3$-8h, OH and $NO_3$
radicals, secondary inorganic aerosols, and reactive VOC species (HCHO) in July 2017 in
China. The model domain, which covers China and surrounding areas in eastern and
southeastern Asia at a 36 km × 36 km horizontal resolution, along with the locations of cities
mentioned in the manuscript, is illustrated in Figure S1.
The simulations included model runs with the following combinations: S11 mechanism
with MEIC inventory (S11-MEIC), S11 with REAS (S11-REAS), CS07 with MEIC (CS07-
MEIC), CS07 with REAS (CS07-REAS), and S18 with REAS (S18-REAS). It should be noted
that while the MEIC emission inventory is for 2017, the most recent year in the REAS
inventory is 2015, which is the one used in the current study. As the MEIC emission inventory
encompasses only emissions from China, emissions from other regions were based on REAS,
even when the simulations were mentioned using the MEIC inventory.



The July anthropogenic emission inventories from MEIC and REAS were processed using
an in-house emission processor, with updated speciation profiles employed to generate CMAQ-
ready VOC emissions (Kang et al., 2022). The speciation profiles for different chemical
mechanisms were derived from the same detailed speciation profiles extracted from the
SPECIATE database. Comprehensive overviews of MEIC and REAS inventories were
provided by Kang et al. (2022). Comparisons of major species, including $NO_x$ ($NO+NO_2$), $SO_2$,
ethene (ETHENE), formaldehyde (HCHO), higher olefins (OLE) (comprising OLE1 and
OLE2, which are lumped alkene species with propylene and trans-2-pentene as representative
compounds), isoprene (ISOPRENE), and monoterpenes (TRP1), are shown in Tables S1-S2
and Figures S2-S3 for municipalities and provinces in July 2017. Note that OLE and TRP1
emissions are for the S11 mechanisms. Emissions for CS07 are similar but some of the species
in OLE and TRP1 are explicit species in S18. There are notable spatial differences in the
weekday emissions of HCHO, ETHENE, OLE, $SO_2$, and $NO_x$ between MEIC and REAS.
Specifically, REAS exhibits higher HCHO emissions in locations like Beijing, Tianjin, Henan,
Shanghai, and Guangzhou than MEIC (Figure S2). In the south of Henan, ethene emissions are
significantly higher in REAS relative to MEIC. The emissions of OLE in REAS are lower than
those in MEIC in Beijing, Tianjin, Shanghai, Guangzhou, Chengdu, and Chongqing. $SO_2$
emissions in MEIC are generally lower than those in REAS, except for Shanghai and
Guangzhou. $NO_x$ emissions differ significantly between MEIC and REAS. In eastern China,
including cities like Shanghai and Guangzhou, $NO_x$ emissions from REAS are typically lower
than those from MEIC, although some areas demonstrate a notable increase. These
discrepancies inevitably affect the accuracy of air pollutant predictions, underscoring the
necessity for a comprehensive assessment of emission inventories in the development of
effective pollution control polices.
Biogenic emissions were generated using the Model of Emissions of Gases and Aerosols
from Nature (MEGAN) version 2.10, which has been observed to emit higher levels of isoprene
and monoterpenes in comparison to anthropogenic sources (Figure S3). Open burning
emissions were produced using FINN inventory from the National Center for Atmospheric
Research (NCAR) (Wiedinmyer et al., 2011). Sea salt and windblown dust emissions were
simulated online using the CMAQ model. Initial and boundary conditions for the model
simulation were generated using CMAQ default profiles. The initial three days of the
simulation serve as a spin-up and are excluded from subsequent analyses.
Meteorological inputs were generated using the Weather Research and Forecasting (WRF)
model version 4.2 for the 36×36 km domain with 44 vertical layers. The initial and boundary



conditions for WRF were derived from the global reanalysis data FNL (available at
https://rda.ucar.edu/datasets/ds083.2/). Further details on the configuration of WRF model can
be found in the work of Kang et al. (2022). WRF-derived meteorological parameters, including
temperature and relative humidity at a height of 2 m above the surface and wind speed and
direction at 10 m, have been validated against observational data from the National Climatic
Data Center (NCDC), demonstrating good performance (Kang et al., 2021).
**2.4 Sensitivity of $O_3$ and related pollutants to emission controls across different**
**mechanisms and inventories**
A large number of sensitivity simulations with systematic reductions in $NO_x$ and VOC
emissions were conducted to explore variations in the sensitivity of $O_3$-8h and related
pollutants to emission reductions across different mechanisms and inventories. Three sets of
simulations were performed for each mechanism/inventory combination considered in this
study (see section 2.3). In the first set of simulations, $NO_x$ emissions were reduced by 20, 40,
50, 60, and 80%, while the emissions of VOCs were maintained at their base-case level. In the
second set of simulations, VOC emissions were reduced by 20, 40, 50, 60, and 80%, while $NO_x$
emissions remained constant. In the third set of simulations, both $NO_x$ and VOC emissions
were reduced by 20-80%.
**3. Results and discussion**
**3.1 Model evaluation of $O_3$ and $PM_{2.5}$ predictions across various mechanisms and**
**inventories**
Observations of $O_3$-8h and $PM_{2.5}$ at a large number of surface monitoring stations nationwide
in July 2017 were obtained from the publication website of the China National Environmental
Monitoring Center (http://www.cnemc.cn). The model performance statistics were evaluated
separately for different regions: the North China Plain (NCP), Yangtze River Delta (YRD),
Central China (Center), Pearl River Delta (PRD), and Sichuan Basin (SCB). Specifically, NCP
includes Beijing, Tianjin, and some cities in Hebei and Shandong provinces; Center includes
cities in the provinces of Henan, Hubei, Hunan, and Jiangxi; YRD includes Shanghai and some
cities in the provinces of Anhui, Jiangsu, and Zhejiang; PRD includes Shenzhen and some
cities in Guangdong province; and SCB includes Chongqing and some cities in Sichuan
province. As shown in Figures S4-S5, the model performance for $O_3$-8h and $PM_{2.5}$ predicted
by different mechanisms and emission inventories in major regions of China exhibits large
variations, due to differences in climate, topography, and emission sources. Overall, the





average values of mean normalized bias (MNB) and mean normalized error (MNE) for $O_3$-8h

predictions across all combinations of mechanisms and inventories are generally within the

model performance criteria suggested by the US EPA in most regions, except for

underprediction in the PRD. Similarly, the model shows good performance for $PM_{2.5}$ in most

areas, except for the PRD region, where the mean fractional biases (MFB) for $PM_{2.5}$ using S18-

REAS, CS07-REAS, and S11-REAS are slightly outside the recommended range (Boylan and

Russell, 2006). The underestimation of $PM_{2.5}$ predictions using REAS in the PRD is likely

related to biases in this inventory specific to this region.

**3.2 Spatial variations in predictions of $O_3$ and related pollutants by different mechanisms**

**and inventories**

Figure 1 shows the spatial distribution of monthly averaged $O_3$-8h concentrations predicted by

S11-MEIC during summer, along with the absolute differences between $O_3$-8h predictions

from S11-MEIC and those from other mechanisms and inventories. Based on S11-MEIC, high

$O_3$-8h levels exceeding 80 ppb occurred in eastern China, especially in Beijing, Tianjin, Hebei,

northern Henan, SCB, and Shanghai. Extremely high $O_3$-8h levels above 100 ppb are also

observed over the Bohai Bay and the Yellow Sea, likely due to regional transport of polluted

air from the continent, reduced $NO_x$ titration with $O_3$, and lower $O_3$ dry deposition velocities

over the ocean. Similar spatial distributions of $O_3$-8h concentrations are also found in the

simulations using other mechanisms and inventories (Figure S6). In comparison, $O_3$-8h

concentrations predicted by S11-REAS are generally 2–7 ppb higher than those by S11-MEIC

in most parts of China, especially in the central region, Zhejiang, and Fujian provinces, but

about 2–7 ppb lower in Beijing, Shanghai, Chengdu, and the PRD region. S18-REAS predicted

significantly lower $O_3$-8h levels, with reductions greater than 7 ppb in regions such as Beijing,

Tianjin, Hebei, Shanghai, SCB, PRD, and other developed areas such as Zhengzhou and Hefei.

In contrast, CS07-MEIC predicted lower $O_3$-8h levels overall, with apparent reductions of ~

6–7 ppb in the SCB, central China, and coastal areas near Shanghai. Similarly, CS07-REAS

also showed generally lower $O_3$-8h concentrations compared to S11-MEIC, particularly in

Chengdu, Luoyang, Shanghai, Guangzhou, Ningbo, Hefei, Nanchang, and the Yellow Sea (~7

ppb lower or more), although some locations exhibit higher $O_3$-8h levels than those predicted

by S11-MEIC. These discrepancies highlight that the accuracy of $O_3$ predictions is sensitive to

variations in photochemical mechanisms and emission inventories. This underscores the

importance of adopting and comparing multiple mechanisms and inventories when developing

region-specific pollution control policies.





Figure 2 illustrates the differences in spatial distribution of monthly averaged SIA

concentrations modeled with different mechanisms and inventories. According to S11-MEIC,

high SIA concentrations are mainly concentrated in NCP and SCB, exceeding 15 μg m$^{-3}$. High

SIA concentrations are also found in Bohai Bay, likely due to long-range transport of polluted

air from land sources. Compared to S11-MEIC, S11-REAS predicts higher SIA concentrations

in most regions, with increases of > 1 μg m$^{-3}$ in most areas, especially in NCP, Henan, and

SCB (higher by about 6 μg m$^{-3}$ or even more). Nationwide, SIA concentrations predicted by

S18-REAS are generally higher than those predicted by S11-MEIC, with increases of up to 6

282       μg m$^{-3}$ or more in the NCP, Central China, SCB, Bohai Bay, and the Yellow Sea. CS07-MEIC

shows similar SIA levels to S11-MEIC, with slightly lower concentrations of ~1 μg m$^{-3}$ in SCB

and Bohai Bay. The spatial differences in SIA predictions between CS07-REAS and S11-

MEIC are similar to those between S11-REAS and S11-MEIC, suggesting that SIA

concentrations from CS07-REAS and S11-REAS are comparable.

**288 3.3 Impacts of mechanisms and inventories on predicted atmospheric oxidation capacity**

Atmospheric oxidation capacity (AOC), which governs the removal rate of primary pollutants

and the production of secondary pollutants (Elshorbany et al., 2009; Prinn, 2003), is primarily

controlled by the hydroxyl (OH) and nitrate (NO$_3$) radicals in the atmosphere (Geyer et al.,

2001; Liu et al., 2022). The spatial differences in OH and NO$_3$ predictions for different

mechanisms and inventories are illustrated in Figures 3-4.

S11-MEIC predicts high OH concentrations exceeding $3.7 \times 10^6$ molecules cm$^{-3}$ (0.15 ppt)

mainly in northern China, including NCP, Inner Mongolia, and some western sites. Compared

to S11-MEIC, S11-REAS predicts lower OH concentrations in NCP, Jiangsu, Shanghai, PRD,

SCB, and other urban nuclei but higher concentrations in rural areas. In contrast, S18-REAS

predicts higher OH concentrations than S11-MEIC in most regions, except in some northern

locations. CS07-MEIC produces elevated OH concentrations in northwestern China, Chengdu,

Chongqing, Shanghai, Hebei, Shandong, and northern Henan, but lower OH levels in other

regions. Similar to S11-REAS, CS07-REAS shows much higher OH levels in northwestern

China but lower OH levels in eastern regions than S11-MEIC. The variability in OH levels can

significantly affect the formation of O$_3$ and other gaseous and particulate pollutants.

The spatial distribution of NO$_3$ concentrations modeled by S11-MEIC exhibits high values

in Xinjiang, Inner Mongolia, NCP, and Bohai Bay, with NO$_3$ concentrations reaching up to 20

306       ppt. Obvious differences in NO$_3$ predictions between S11-REAS and S11-MEIC are primarily



observed in Xinjiang and Inner Mongolia, with smaller differences in other regions. S18-REAS
generally predicted higher NO₃ concentrations than S11-MEIC, especially in Xinjiang, Inner
Mongolia, Bohai Bay, Yellow Sea, and Taiwan Strait, by ~6 ppt or more. In contrast, CS07-
MEIC consistently predicts lower NO₃ levels than S11-MEIC. CS07-REAS predicted lower
NO₃ concentrations in the eastern regions but higher concentrations in parts of Xinjiang
compared to S11-MEIC.

**3.4 Impacts of mechanisms and inventories on HCHO prediction**
Spatial variations in AOC, as reflected in OH and NO₃ predictions from different mechanisms
and inventories, imply that photochemical formation and loss rates for air toxics may also vary
depending on the mechanism and inventory used. To explore how these variations affect the
modeling results for gaseous pollutants, we examined HCHO, one of the most important
gaseous air toxics with both primary and secondary sources. It is also a significant contributor
to O₃ formation and OH production, thus playing a crucial role in tropospheric photochemistry
(Wang et al., 2017; Zhang et al., 2013).

HCHO concentrations predicted by S11-MEIC are generally high in the SCB and eastern

China, areas with significant biogenic VOC emissions (Kang et al., 2023), particularly in
Chengdu, Shanghai, and Changsha, where HCHO concentrations reach or exceed 7 ppb (Figure
5). This suggests that a significant fraction of the HCHO is due to secondary formation. HCHO
levels predicted by S11-REAS are similar to those predicted by S11-MEIC, with only minor
differences of less than 1 ppb. In contrast, S18-REAS predicts significantly lower HCHO levels
than S11-MEIC, with differences > 1 ppb, especially in SCB, NCP, PRD, YRD, and central
China. CS07-MEIC and CS07-REAS predict higher HCHO levels than S11-MEIC, especially
in SCB and eastern China, with differences exceeding 1 ppb. These results indicate that HCHO
predictions are more strongly influenced by photochemical mechanisms than by uncertainties
in emission inventories, due to its secondary formation.

**3.5 Impacts of mechanisms and inventories on the sensitivity of O₃ and related pollutants**
**to emission controls**
Figures 6-10 and S6-S15 display the fractional changes in predictions of O₃ and related
pollutants due to NOₓ and VOC reductions using various mechanisms and inventories.
**3.5.1 Impacts of mechanisms and inventories on O₃ sensitivity to emission controls**
For all mechanisms and inventories, O₃-8h concentrations consistently decrease with
reductions in NOₓ, VOCs, or both in all five cities during summer. The efficiency of emission



controls improves with increasingly higher emission reductions of $NO_x$, VOCs. These trends
are consistent with the findings of Kang et al. (2021), suggesting that the mechanisms and
inventories do not affect the trend of change in $O_3$, but do affect the magnitude of the change.
These results also show that during summer, the sensitivity of $O_3$ formation to $NO_x$ and VOCs
is in transition or $NO_x$-limited regimes. In Beijing, when $NO_x$ emissions are reduced, CS07-
MEIC and CS07-REAS show the largest $O_3$-8h reductions compared to other mechanisms and
inventories, while S18-REAS exhibits the smallest changes, especially with larger $NO_x$
reductions. In Shanghai, the largest $O_3$-8h reductions due to $NO_x$ controls are observed with
CS07-REAS, followed by S11-REAS, while S18-REAS again shows the smallest changes. In
Changsha, there are no significant differences among mechanisms and inventories in the
sensitivity tests. In Shenzhen, S11-REAS shows the greatest decreases in $O_3$-8h, followed by
CS07-REAS and S18-REAS, while S11-MEIC and CS07-MEIC show the smallest changes.
Except for S18-REAS, predictions from other mechanisms and inventories are similar in
Chongqing.
When only VOC emissions are reduced in Beijing, S11-MEIC predicts the largest $O_3$-8h
reductions, while CS07-REAS predicts the smallest changes. Similar patterns are found in
Shanghai and Changsha, with larger differences among different mechanisms and inventories
in Shanghai. In Shenzhen, S11-MEIC also exhibits the largest $O_3$-8h decreases, followed by
CS07-MEIC, while CS07-REAS, S11-REAS, and S18-REAS show smaller changes. In
Chongqing, S18-REAS predicts the largest changes and CS07-MEIC the smallest, although
the differences among these scenarios are not substantial. When both $NO_x$ and VOC emissions
are reduced, the predicted change rates of $O_3$-8h do not vary much across different mechanisms
and emission inventories for all five cities. Additionally, the sensitivity tests for Shenzhen, as
presented in Figure 6, suggest that differences in emission inventories may have a greater
impact on emission control outcomes than differences in chemical mechanisms.
Figure S6 illustrates the national-scale relative changes in $O_3$-8h due to $NO_x$ emission
reductions evaluated using different mechanisms and inventories. The results are consistent
with those shown in Figure 6, with the greatest $O_3$-8h reductions occurring in the SCB, Central,
and YRD. For all mechanisms and inventories, increasingly higher $NO_x$ reductions generally
lead to increasingly lower $O_3$ in summer, with relative changes in $O_3$-8h ranging from
approximately 5% to 60% as $NO_x$ reductions increase from 20% to 80%. In comparison, the
$O_3$-8h reductions predicted by S18-REAS are less pronounced than other mechanisms and
inventories in most areas. Similar to $NO_x$ reductions, higher VOC reductions typically result
in larger $O_3$ reductions in summer, with significant reductions observed in NCP, Chengdu,



PRD, Shanghai, and Bohai Bay. The relative changes in $O_3$-8h increase from 5% to 40% as
VOC reductions increase from 20% to 80% in these areas (Figure S7). The comparison
between Figure S6 and Figure S7 suggests that $NO_x$ reduction tends to be more effective in
controlling $O_3$ pollution in non-VOC-limited regions than VOC reduction, given the same level
of emission reduction.
**3.5.2 Impacts of mechanisms and inventories on SIA sensitivity to emission controls**
Reducing $NO_x$ emissions typically leads to decreasing SIA levels across the five cities, with
the efficiency of reductions improving as $NO_x$ reduction increases. As illustrated in Figure 7,
the predicted changes in SIA are quite similar for different mechanisms and inventories.
However, VOC controls do not always effectively reduce SIA concentrations, likely
attributable to an increase in $NO_3$ radicals, as shown in Figure 9. In Beijing, changes in SIA in
response to VOC reductions vary with mechanisms and inventories. For S11-MEIC and CS07-
MEIC, SIA concentration first increases and then decreases with decreasing VOCs. S18-REAS
shows a general decrease in SIA concentrations with decreasing VOCs. In contrast, CS07-
REAS and S11-REAS predict an increase in SIA concentrations with VOC reductions, with
the rate of SIA growth initially increasing and then decreasing as VOC emission decreases. In
Shanghai, VOC reductions generally increase SIA concentrations across all mechanisms and
inventories, especially for CS07-REAS and S11-REAS, which show that SIA concentrations
consistently increase as VOCs are progressively reduced. In Changsha, SIA concentrations also
consistently increase with increasing VOC reductions for all mechanisms and inventories. In
Shenzhen, SIA concentrations increase with VOC reductions; however, the changes are more
pronounced for S11-REAS compared to the less significant variations observed for CS07-
MEIC. Except for S18-REAS, Chongqing shows an increase in SIA concentrations with VOC
reductions, particularly for CS07-REAS. Overall, simultaneous reductions in $NO_x$ and VOCs
decrease SIA levels in most cities, similar to the effects of $NO_x$ controls alone.
Figure S8 shows the relative changes in SIA due to $NO_x$ controls on a national scale, using
different mechanisms and inventories. The stepwise reductions in $NO_x$ emissions lead to an
overall decrease in SIA concentrations, with the most pronounced reductions observed in the
NCP, SCB, Jiangsu, Bohai Bay, and Yellow Sea. The decreases in SIA increase from ~5% to
60% as $NO_x$ reduction increases from 20% to 80%. On the contrary, VOC reduction causes an
overall increase in SIA concentrations (Figure S9), indicating that VOC controls are less
effective than $NO_x$ controls in reducing SIA levels during summer. Notably, the largest
increases in SIA are seen in SCB, Central, and YRD, where SIA increases grow from ~ 6% to



30% as VOC reduction increases from 20% to 80%. In comparison, S18-REAS shows
relatively smaller changes in SIA, with some areas in NCP showing obvious decreases in SIA
concentrations.

**3.5.3 Impacts of mechanisms and inventories on OH and NO$_3$ sensitivity**
As shown in Figures 8-9, the effects of NO$_x$ reductions on OH concentrations vary depending
on the mechanism and inventory used. In most regions, OH production decreases with reducing
NO$_x$ emissions, likely due to the decrease in O$_3$ concentrations, as O$_3$ photolysis in the presence
of water vapor is a significant source of atmospheric OH (Seinfeld and Pandis, 2016). Except
for Shenzhen, CS07-MEIC and CS07-REAS predict the largest decreases in OH due to NO$_x$
control in other cities, while S18-REAS predicts the smallest OH change rates. In Shenzhen,
OH levels predicted by S11-MEIC and CS07-MEIC initially increase and then decrease as NO$_x$
emissions decrease, while those predicted by S18-REAS, CS07-REAS, and S11-REAS exhibit
a consistent decrease in OH levels. When only VOCs are reduced, the changes in OH vary
across different mechanisms and inventories. In Beijing, except for S18-REAS, OH
concentration initially increases and then decreases with decreasing VOC emissions. In
Shanghai, OH concentrations from CS07-REAS and S11-REAS basically increase with
decreasing VOCs, while those from CS07-MEIC and S11-MEIC show an initial increase
followed by a decrease. S18-REAS predicts a general decrease of OH with decreasing VOCs
in Beijing and Shanghai. In Changsha, OH concentrations consistently increase with reduced
VOCs across all mechanisms and inventories, with CS07-REAS showing a rapid increase of
up to 75%. In Shenzhen, OH levels from CS07-REAS, S11-REAS, and S18-REAS increase
with decreasing VOCs, while those from S11-MEIC and CS07-MEIC decrease. In Chongqing,
all mechanisms and inventories except S18-REAS predict an increase in OH concentrations
with decreasing VOCs, with CS07-MEIC showing a significant increase of up to 75%. When
both NO$_x$ and VOC emissions are reduced, OH concentrations generally increase for all
mechanisms and inventories, although the relative changes are relatively small ($< 25\%$).

Figures S10-S11 display the spatial distribution of OH changes due to incremental emission

controls across different mechanisms and inventories. There is a nationwide decrease in OH
concentrations due to NO$_x$ reductions, with more pronounced decreases in central and eastern
China, where OH levels drop by $\sim$ 20% to 80% as NO$_x$ reduction increases from 20% to 80%.
Instead, VOC reductions generally lead to increased OH concentrations across most of China,
significantly in some regions with large vegetation cover (Kang et al., 2023), such as central
and southeastern China. In these areas, OH levels increase by 20~200% in response to stepwise



VOC reductions. This increase occurs because high atmospheric VOC concentrations in rural
and vegetation-rich areas react with OH radicals to produce $RO_2$ and $HO_2$, depleting OH, thus
reducing VOCs in these areas increases OH levels. However, in some urban centers, such as
Beijing and Shanghai, changes in OH levels due to VOC controls depend on the extent of
reduction and the choice of mechanism and inventory.
Similar to OH, $NO_3$ levels predicted by different mechanisms and inventories basically
decrease with decreasing $NO_x$ emissions, likely related to the decline of $O_3$ concentrations since
$NO_3$ radicals are predominantly formed by the reaction of $NO_2$ with $O_3$ (Geyer et al., 2001). In
general, the change rates of $NO_3$ due to $NO_x$ reductions are higher for CS07-MEIC and CS07-
REAS, while lower for S18-REAS. When VOC emissions are reduced, $NO_3$ production
increases, except for S18-REAS. In Changsha, a particularly dramatic increase of over 200%
and 150% is observed for CS07-REAS and CS07-MEIC, respectively, when VOC emissions
are cut by 80%. This increase in $NO_3$ radicals could be attributed to a decline in the rapid
reaction of $NO_3$ with unsaturated hydrocarbons. However, for S18-REAS, $NO_3$ concentrations
decrease with decreasing VOCs in Beijing, Shanghai, and Chongqing. Unlike OH,
simultaneous reductions in $NO_x$ and VOCs result in lower $NO_3$ levels across all mechanisms
and inventories.
Figures S12-13 illustrate the spatial variation in $NO_3$ changes due to systematic emission
reductions for all mechanisms and inventories. $NO_3$ radicals show a decreasing trend,
particularly in eastern China, where reductions in $NO_3$ levels vary from 40% to 100%. Except
for S18-REAS, other mechanisms and inventories generally predict a nationwide increase in
$NO_3$ radicals with reduced VOCs, particularly in central and southeastern China, where $NO_3$
levels increase by ~ 30 – 300% or even more with increasing VOC reductions. S18-REAS
predicts a decrease in $NO_3$ levels with reduced VOCs in some megacities such as Beijing,
Shanghai, Chengdu, and Chongqing, while $NO_3$ levels increase elsewhere.

**3.5.4 Impacts of mechanisms and inventories on the sensitivity of secondary gaseous**
**pollutants to emission controls**
As shown in Figure 10, HCHO concentrations decrease with decreasing $NO_x$ emissions, likely
due to decreased AOC, as evidenced by decreased OH and $NO_3$ levels. This indicates that
secondary formation from the oxidation of atmospheric VOCs is the dominant source of HCHO,
in accordance with previous studies (Wang et al., 2017; Yang et al., 2019; Zhang et al., 2013).
In addition, the most significant decreases in HCHO levels due to $NO_x$ reductions are observed
for CS07-REAS in all five cities, particularly in Changsha and Chongqing, where HCHO levels





drop by about 40%. When VOCs are reduced alone, HCHO levels decrease linearly across all
mechanisms and inventories, a trend similar to the simultaneous reduction of $NO_x$ and VOCs.
Variations in HCHO changes across different mechanisms and inventories suggest the need to
evaluate different mechanisms and inventories when formulating regional emission control
policies for carbonyl pollution.
Figures S14- S15 illustrate the spatial distribution of relative changes in HCHO due to $NO_x$
and VOC controls using different mechanisms and inventories. VOC and $NO_x$ controls lead to
reductions in HCHO concentration, especially in eastern China. However, VOC controls
generally leads to larger reductions in HCHO concentrations (~10–80%) than $NO_x$ controls (~
4 – 40%), as shown in Figures S14-15 and 10. This suggests that VOC controls are also
essential and more effective for reducing secondary gaseous organic pollutants in the
atmosphere.

**4. Conclusions**

This study utilized the CMAQ model to evaluate the impacts of different mechanisms and
inventories on the prediction of $O_3$ and other air pollutants. It also examined how these
mechanisms and inventories affect the sensitivity of $O_3$ and related species to emission
reductions. For maximum daily average 8-hour $O_3$ ($O_3$-8h), relative reductions predicted using
different emission inventory and mechanism combinations are consistent for up to 80% $NO_x$
or VOC reductions, with maximum differences of approximately 15%. For secondary
inorganic aerosols (SIA), while the predicted relative changes in their daily average
concentrations due to $NO_x$ reductions are quite similar, very large differences of up to 30%
occur for VOC reductions. Sometimes even the direction of change (i.e., increase or decrease)
is different. For the oxidants OH and $NO_3$ radicals, the uncertainties in the relative changes due
to emission changes are even larger among different inventory-mechanism combinations,
sometimes by as much as 200%. Our results suggest that while the $O_3$-8h responses to emission
changes are not sensitive to the choice of chemical mechanism and emission inventories, using
a single model and mechanism to evaluate the effectiveness of emission controls on SIA and
atmospheric oxidation capacity may have large errors. For these species, the evaluation of the
control strategies may require an ensemble approach with multiple inventories and mechanisms.

**Data availability**. The dataset for this paper is available upon request from the corresponding
author (qying@ust.hk).



**Author contributions.** QY designed the study. QY and MJK developed the CMAQ model. MJK conducted the simulations, analyzed the data, and wrote the manuscript. All coauthors contributed to the discussion and revision of the paper.

**Competing interests**. The authors declare that they have no conflict of interest.

**Acknowledgments**

This work was partly supported by the National Natural Science Foundation of China (No. 42307142).



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



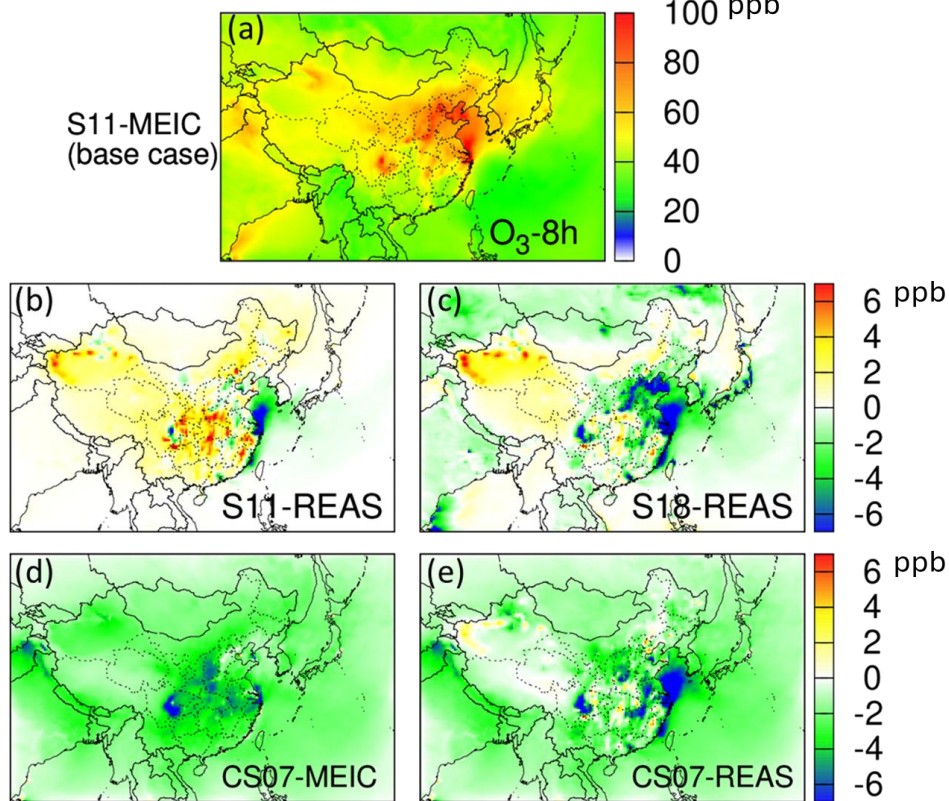

**Figure 1.** Predicted monthly averages of MDA8 $O_3$ ($O_3$-8h) concentrations in July 2017 using (a) the S11 mechanism and MEIC emission inventory (base case), and the differences between the base case and cases using alternative photochemical mechanisms and emission inventories (alternative case – base case): (b) S11 and REAS, (c) S18 and REAS, (d) CS07 and MEIC, and (e) CS07 and REAS inventories. Units are ppb.

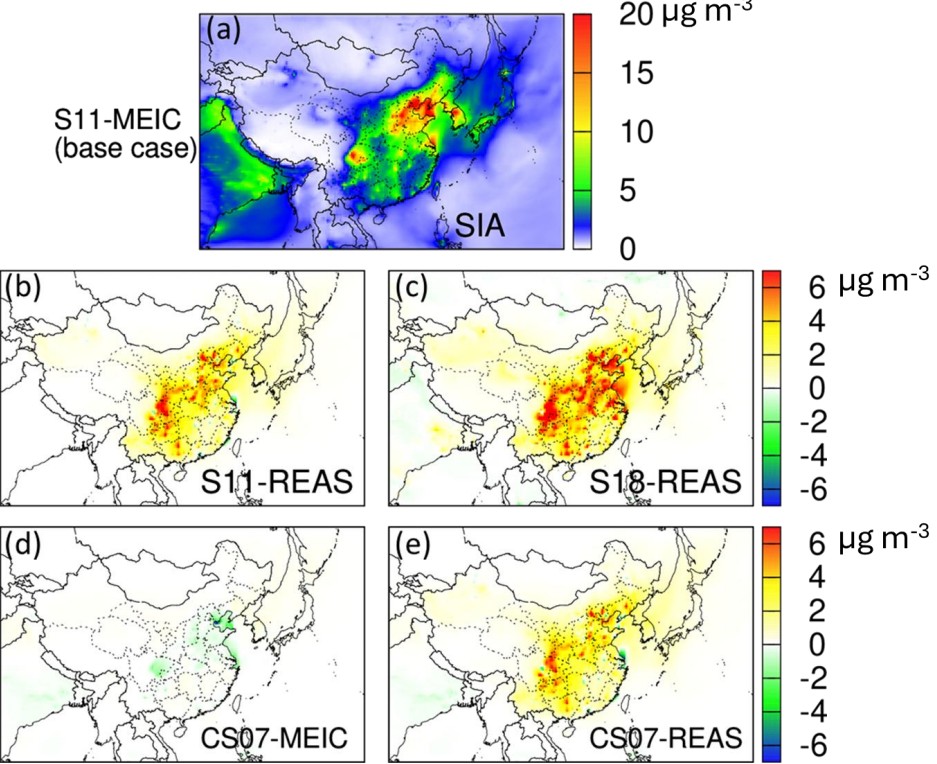

**Figure 2.** Predicted monthly averages of secondary inorganic aerosol (the sum of nitrate, sulfate and ammonium ion, SIA) concentrations in July 2017 using (a) the S11 mechanism and MEIC emission inventory (base case), and the differences between base case and cases using other photochemical mechanisms and emission inventories (alternative case – base case): (b) S11 and REAS, (c) S18 and REAS, (d) CS07 and MEIC, and (e) CS07 and REAS. Units are $\mu g\ m^{-3}$.



727

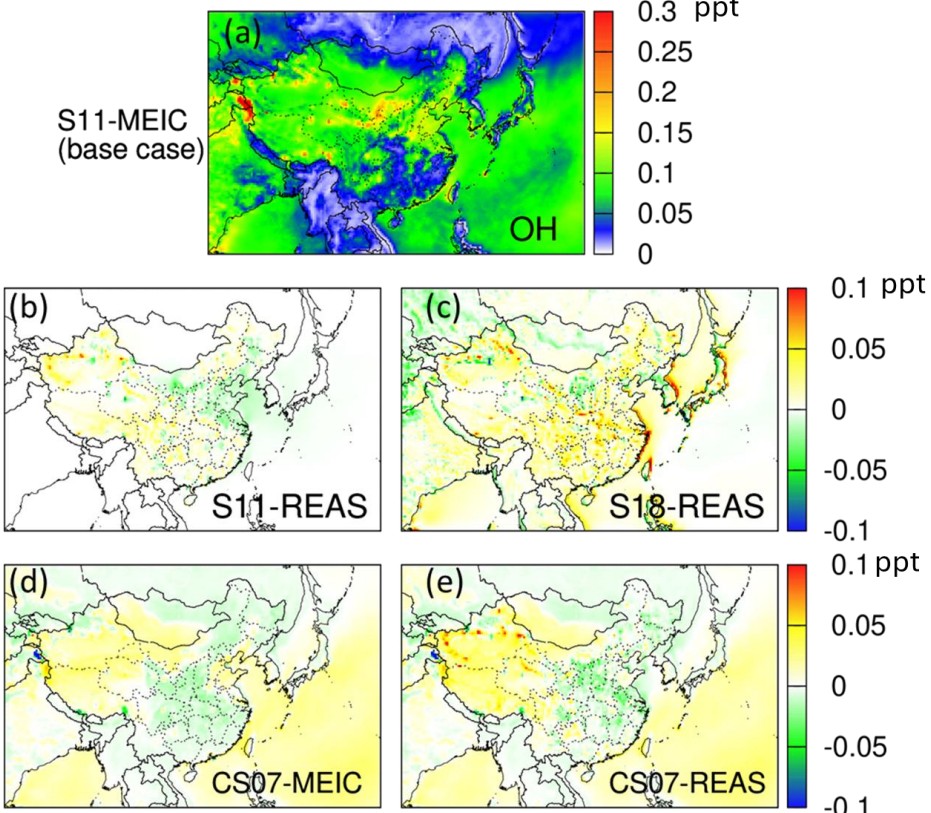

728
729

**Figure 3.** Predicted monthly averages of OH radical concentrations in July 2017 using (a) the S11 mechanism and MEIC emission inventory (base case), and the differences between the base case and cases using other photochemical mechanisms and emission inventories (alternative case – base case): (b) S11 and REAS, (c) S18 and REAS, (d) CS07 and MEIC, and (e) CS07 and REAS. Units are ppt. (0.1 ppt ~ $2.46 \times 10^6$ molec cm$^{-3}$)

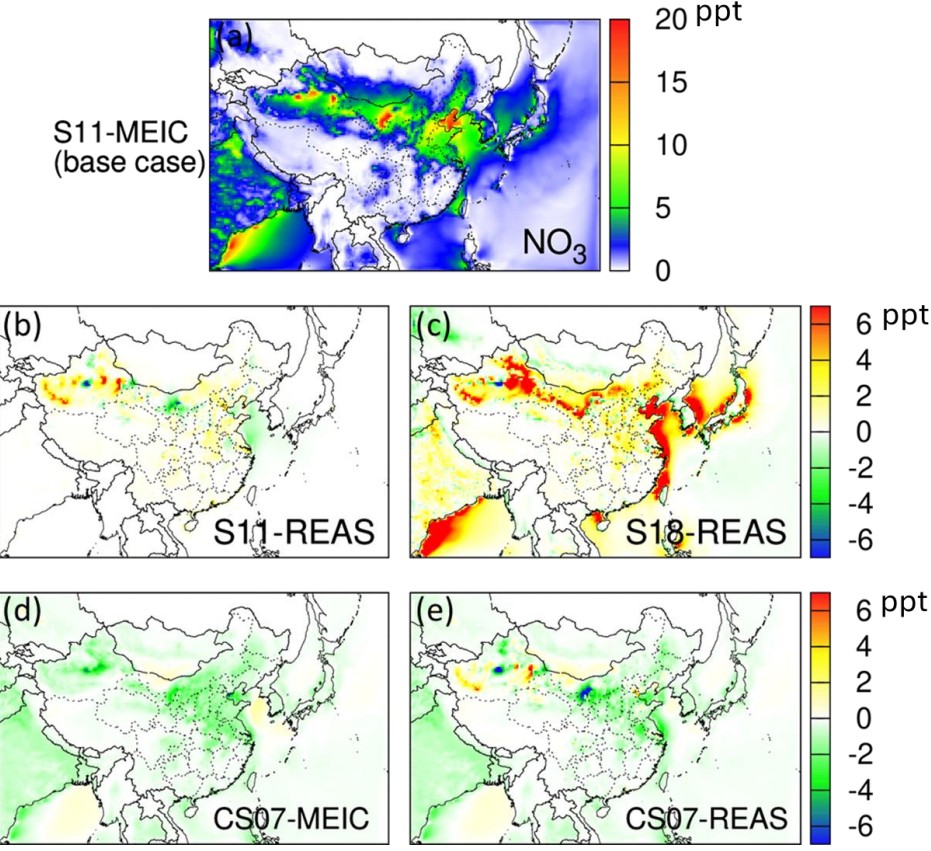

**Figure 4.** Predicted monthly averages of $NO_3$ radical concentrations in July 2017 using (a) the S11 mechanism and MEIC emission inventory (base case), and the differences between the base case and cases using alternative photochemical mechanisms and emission inventories (alternative case – base case): (b) S11 and REAS, (c) S18 and REAS, (d) CS07 and MEIC, and (e) CS07 and REAS. Units are in ppt. (0.1 ppt ~ $2.46 \times 10^6$ molec cm$^{-3}$)

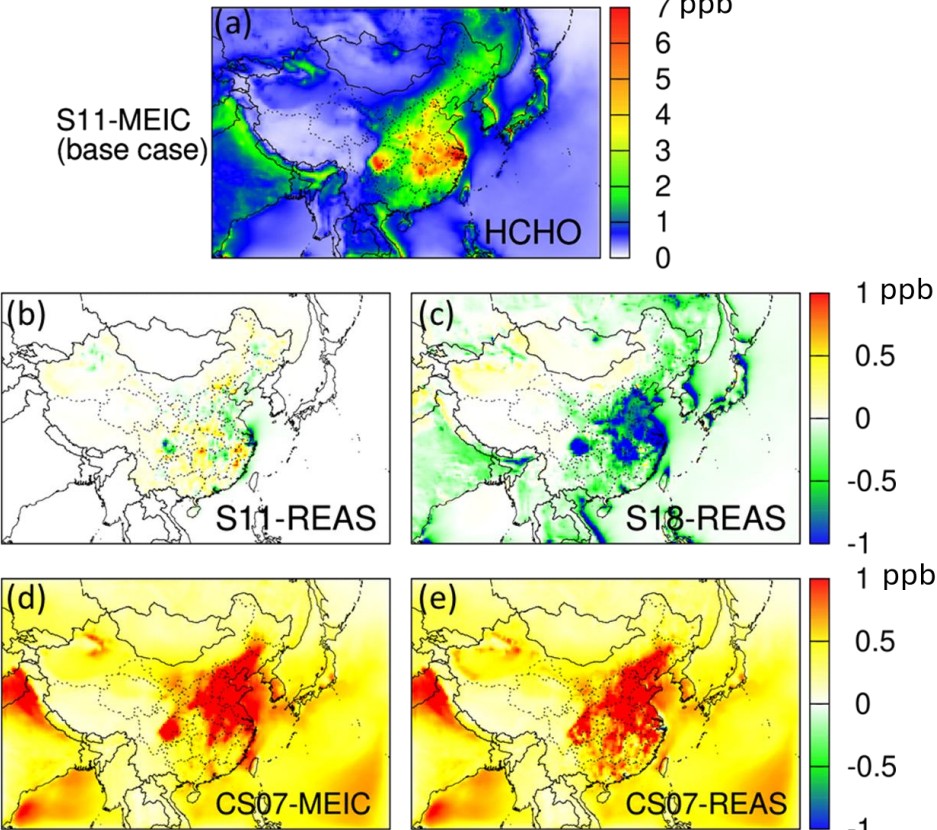

**Figure 5.** Predicted monthly averages of HCHO concentrations in July 2017 using (a) the S11 mechanism and MEIC emission inventory (base case), and the differences between the base case and cases using alternative photochemical mechanisms and emission inventories (alternative case – base case): (b) S11 and REAS, (c) S18 and REAS, (d) CS07 and MEIC, and (e) CS07 and REAS. Units are ppb.


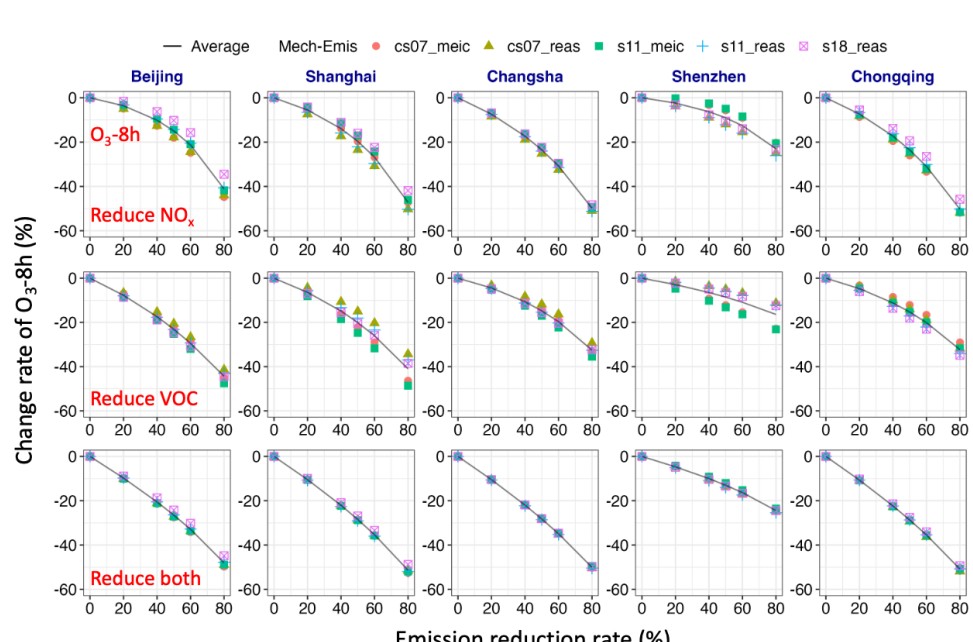

**Figure 6.** Predicted changes of monthly average MDA8 O$_3$ (O$_3$-8h) concentrations in July 2017
due to reductions of NO$_x$ only (first row), VOCs only (second row), and NO$_x$ and VOCs (third
row) using different photochemical mechanisms and emission inventories. The black line
represents the average change across all mechanisms and inventories at different levels of
emission reductions.

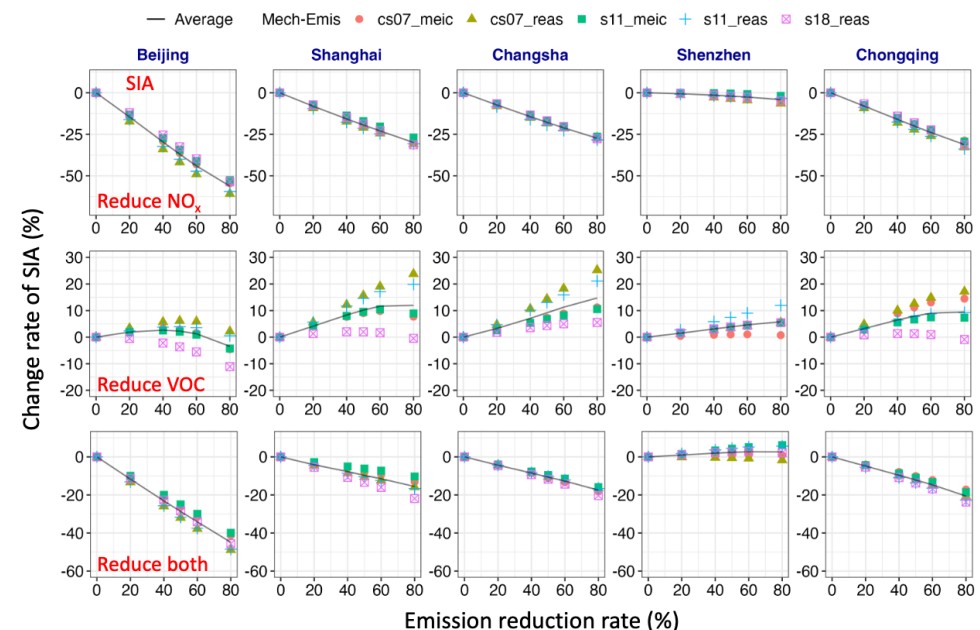

**Figure 7.** Predicted changes of monthly average secondary inorganic aerosol (SIA) concentrations in July 2017 due to reductions of $NO_x$ only (first row), VOCs only (second row), and $NO_x$ and VOCs (third row) using different photochemical mechanisms and emission inventories. The black line represents the average change across all mechanisms and inventories at different levels of emission reductions.




















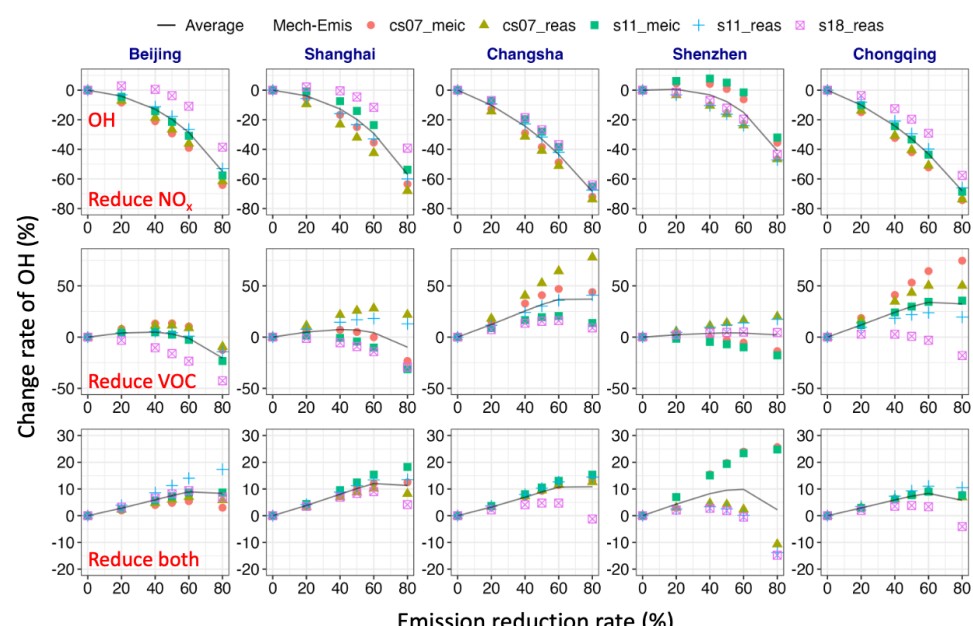

**Figure 8.** Predicted changes of monthly average OH radical concentrations in July 2017 due to reductions of NO$_x$ only (first row), VOCs only (second row), and NO$_x$ and VOCs (third row) using different photochemical mechanisms and emission inventories. The black line represents the average change across all mechanisms and inventories at different levels of emission reductions.











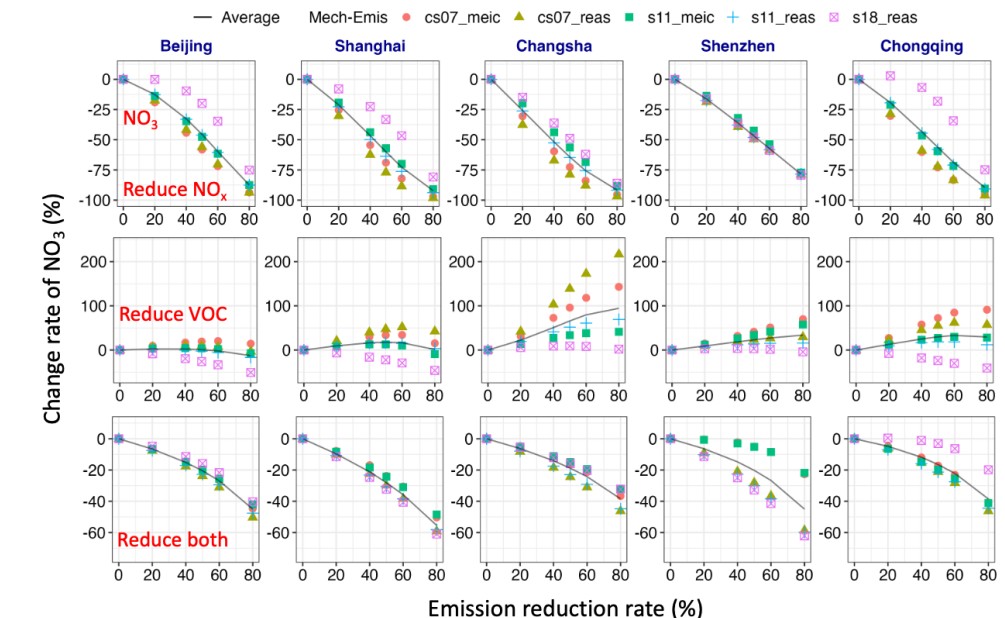

**Figure 9.** Predicted changes of monthly average NO₃ radical concentrations in July 2017 due to reductions of NO$_x$ only (first row), VOCs only (second row), and NO$_x$ and VOCs (third row) using different photochemical mechanisms and emission inventories. The black line represents the average change across all mechanisms and inventories at different levels of emission reductions.



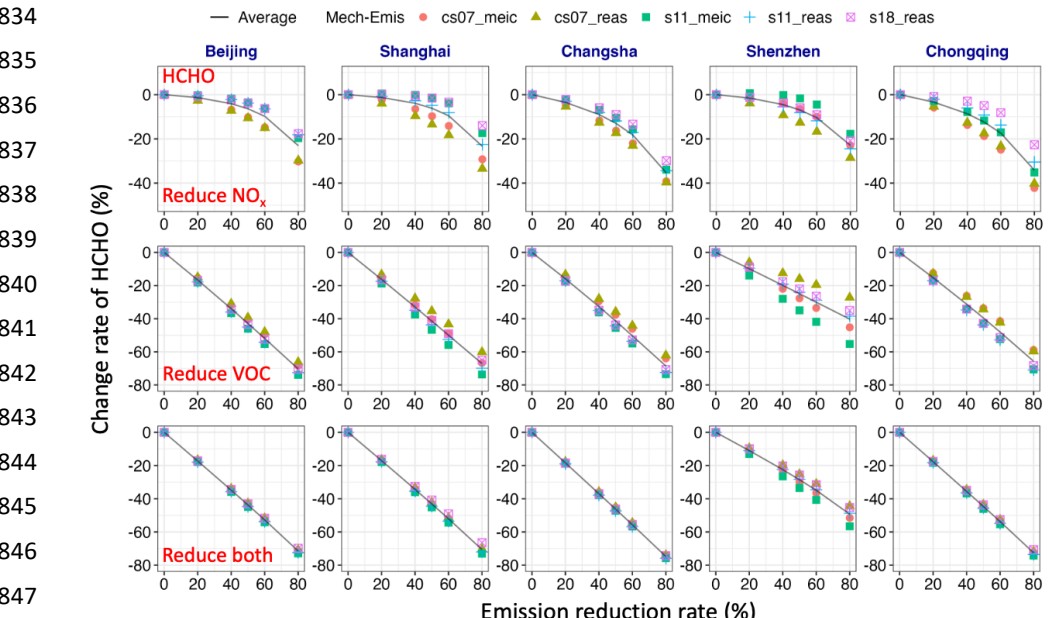

**Figure 10.** Predicted changes of monthly average HCHO concentrations in July 2017 due to reductions of NO$_x$ only (first row), VOCs only (second row), and NO$_x$ and VOCs (third row) using different photochemical mechanisms and emission inventories. The black line represents the average change across all mechanisms and inventories at different levels of emission reductions.