# Peer review of "Effectiveness of Emission Controls on Atmospheric Oxidation"

_EGUsphere, 2025_

## Author Comment (AC2)

**Response to reviewers' comments**

**Response to RC2:**

This study investigated the uncertainties of emission controls on atmospheric oxidation and air pollutant concentrations by comparing different chemical mechanisms and inventories using an advanced CMAQ model. The manuscript is well-organized, clearly written, and the conclusions are well-supported by the presented data. These findings are timely and provide valuable insights for designing effective region-specific emission control strategies. I recommend this manuscript for publication in ACP after addressing the following minor revisions.

We appreciate the reviewer's feedback on the manuscript, and we carefully reviewed the comments and addressed each individually below, highlighting changes made in the revised manuscript.

Minor suggestions:

Line 28: Since the full name of SIA has already been introduced, the abbreviation can be used directly here.

SIA (Secondary Inorganic Aerosol) is spelled as suggested.

Lines 181-183: The statement "Note that OLE and TRP1 emissions are for the S11 mechanisms" is somewhat unclear. The authors should rephrase this sentence to improve clarity.

The sentence has been changed to "Note that OLE and TRP1 emissions are from the S11 mechanisms".

Lines 241-243: The criteria suggested by the US EPA are better to be stated in the text for clarity.

The criteria for O3 are ±0.15 for MNB and 0.3 for MNE. The sentence has been changed to "...within the suggested model performance criteria (MNB≤±0.15 and MNE≤0.3) by Emery et al. (2017) in most regions."

Lines 246-247: The current justification appears weak. The authors should revise this sentence.

The sentence in question is a bit vague, it is revised to "The underestimation of PM2.5 predictions using REAS in the PRD, as shown in Figure S5, is likely related to biases in this inventory specific to this region."

Lines 257-258: The authors attribute high O3 levels over water bodies to lower O3 dry deposition velocities over the ocean. Is there existing literature supporting this argument? A reference is needed here.

References are added to the sentence.

Line 274: The full term for SIA should be provided upon its first appearance in the main text. In addition, Figure 7 should explicitly list the SIA components for clarity.

The SIA is spelled out in the revised manuscript. The caption of Figure 7 is updated.

Lines 494 and 497: Since O3-8h and SIA have been previously defined, the abbreviation can be used directly without reintroducing the full terms.

This is corrected.

References

Emery, C., Liu, Z., Russell, A. G., Odman, M. T., Yarwood, G., and Kumar, N.: Recommendations on statistics and benchmarks to assess photochemical model performance, Journal of the Air & Waste Management Association, 67, 582–598, https://doi.org/10.1080/10962247.2016.1265027, 2017.

---

## Author Response (AR1)

**Response to reviewers' comments**

**Response to RC1:**
Overview comment

This manuscript provides the similarities and/or differences of responses due to the choice of chemical mechanism and emission inventories. I will agree with the final remark, "the evaluation of the control strategies may require an ensemble approach with multiple inventories and mechanisms," for a better understanding of the model and clear guidance for policymakers. I could follow most parts of this manuscript; however, I am afraid that the manuscript may be hard to understand for those outside of modeling researcher. The fundamental revision, especially in the introduction and methodology, will be required. Please address the following comments.

We appreciate the reviewer's feedback on the manuscript, and we carefully reviewed the comments and addressed each individually below, highlighting changes made to the revised manuscript.

Major comments

Explanation of SAPRC mechanism: Lines 79-81, and even reading after Line 127-144 (Section 2.1), it was very hard to understand the SAPRC mechanism, especially for an unfamiliar researcher in modeling. Please rewrite and reorganize these parts to provide a clear introduction and explanation of the method.

We sincerely appreciate the reviewer's feedback. We recognize that the original explanation of the SAPRC mechanism may not have been easy for readers without prior expertise in atmospheric chemistry modeling. Given your suggestion, we added the following discussion to improve readability for researchers unfamiliar with atmospheric chemical mechanisms:

"The SAPRC mechanism is a widely used photochemical mechanism that represents complex atmospheric reactions in computationally tractable forms. Instead of tracking the oxidation of individual precursor organic compounds and their reaction products explicitly, the SAPRC is a lumped-molecule chemical mechanism that groups structurally similar VOC species (e.g., alkanes, akenes, and aromatics) into several groups of lumped model species. Some important species, such as isoprene and formaldehyde, are represented explicitly. The reactions of each lumped model species with common oxidants (OH, $NO_3$ and $O_3$) are derived based on the reactions of individual species within that group, which are generated automatically using a mechanism generator (Carter et al., 2025). One of the complexities in representing VOC reactions is the intermediate oxidation products and radical species. In the SAPRC mechanism, the radicals such as peroxyl radicals and intermediate products such as organic nitrates are represented by a group of common species in order to reduce the number of reactions and model species. A brief comparison of the three SAPRC mechanisms are summarized in Table S1."

The above discussion is included in the revised manuscript on pages 5-6, lines 127-139.

Configuration of experiments: Due to the inconsistency of the targeted year of the emission inventory, a large difference in the comparison is natural. What is the motivation for using different years? Because of this inconsistency of the year in the emission inventory and observation, how do we understand the result of the modeling evaluation (Section 3.1)? Unfortunately, without understanding this modeling comparison design, I cannot go on to read after Section 3.2.

As mentioned by the reviewer, the large differences between REAS (base year 2015) and MEIC (base year 2017) are expected because they represent emissions from different years and different activity and emission factors used to estimate the emissions. In general, the REAS emissions are lower than the MEIC emissions (see Figure S2). If the REAS emissions were adjusted to account for the emission reductions from 2015 to 2017, the difference between REAS and MEIC would be even larger. However, the changes are likely incremental because no major policy shifts occurred during this time. Since it is unclear whether the reduced REAS emissions would better reflect the actual emissions in 2017, we chose not to adjust the REAS emissions to provide a conservative estimation of the difference between the two emission inventories.

As discussed in Section 3.1, the model performance results show that the two emission inventories lead to similar performance. Based on the model performance criteria recommended by Emery et al. (2017), both emissions can lead to acceptable $O_3$ and $PM_{2.5}$ model performance in most parts of China (Figures S4 and S5). For the same emission inventory, the model performance of $O_3$ and $PM_{2.5}$ is also similar. The model performance evaluation in Section 3.1 shows that different combinations of emission inventories and mechanisms lead to acceptable model performance. This suggests that uncertainties in emission estimations do not significantly affect the model's ability to predict $O_3$ and $PM_{2.5}$ concentrations. This further supports our decision not to adjust the REAS emissions. However, there are still large differences in the model predictions, and it is unclear how the selection of mechanisms and emissions will affect the assessment of the effectiveness of emission controls.

Specific comments

Line 100 and Lines 147-149: My simple question is why EDGAR was not used in this study. Please clarify the reason for the selection of MEIC and REAS in this study. From the website indicated in Line 148, REAS seems to be updated to 3.2.1. Why was the older version applied even though the release of the updated version?

Previous studies have shown that using EDGAR emissions do not lead to better model performance of $O_3$ and $PM_{2.5}$ than the MEIC and REAS predictions (Hu et al., 2017). Additionally, including another inventory would require at least 48 additional simulations, which are computationally intensive.  In addition, we have used REAS 3.1 in several of our previous studies (Kang et al., 2021a, 2022a, b, 2023), which have confirmed that REAS 3.1 could lead acceptable

model performance. Moreover, the model year 2017 is not far from the base year of REAS 3.1 (year 2015). Thus, we did not update the emissions to REAS 3.2.1 in this study. However, in future studies, we will update the REAS emissions to 3.2.1.

Line 161: The reason why July was analyzed in this study was unclear. Please clearly introduce this reason.

July is widely used in modeling studies to represent conditions of a typical summer month (Kang et al., 2021a).

Line 167: Why was S18-MEIC not conducted?

The MEIC only has five emission sectors, making it difficult to re-speciate the emissions for the S18 mechanism. Specifically, solvent utilization emissions are not represented in the public version of MEIC as a separate sector (Wang et al., 2018). Since solvent utilization accounts for a significant fraction of VOC emissions in urban areas and has very different emission characteristics than fuel combustion sources, re-speciating the five-sector MEIC VOC emissions will be inaccurate.

The above discussion is included in the revised manuscript on page 7, lines 180-186.

Lines 172-195: Again, I wonder what the intention is of this discussion under the configuration of a different year's emission inventory.

Even though the two emissions are different, it is still necessary to understand the magnitude of the differences. Furthermore, the discussion of quantitative difference in the emissions of individual species is necessary to explain the difference in the predicted concentrations.

Line 223: It is better to explicitly state "20, 40, 50, 60, and 80%".

The sentence is revised according to the suggestion.

Lines 251-253 (Figure 1): I understand that the emission outside China is unified by REAS. So, what stands for a larger difference over Mongolia, eastern Russia, Japan, and eastern India shown in Fig. 1(c)? Is this due to the different treatment in S11 and S18? Anyway, a detailed introduction to understand the differences in the SAPRC scheme is insufficient, and we do not follow these differences.

The difference in Fig. 1(c) for other countries is mainly due to the difference between S11 and S18. Predicted $O_3$ concentrations by S18 are lower than those by S11, especially in urban areas (e.g., in Ulaanbaatar in Mongolia). This is consistent with the box-model simulation results reported by Carter (2020). In Fig. 1(a), the difference of the monthly $O_3$ in other countries is negligible when the same chemical mechanism is used, which further confirms our conclusion.

This clarification is included in the revised manuscript on page 10, lines 293-298.

Figures 3 and 4: The above comment could also be repeated for radicals of OH and NO3. A larger difference outside China was obvious in Fig. 3(c) and Fig. 4(c). However, there is no mention of these differences in Section 3.3. Please clarify.

Yes, the differences are mainly due to the different chemical mechanisms. S18 is slightly higher or lower than S11 in predicted OH radicals, which is consistent with the box-model simulation results reported by Carter (2020). Carter's box-model simulation does directly report the difference of $NO_3$ radicals between S11 and S18. However, the $NO_2$ concentration predicted by S18 is significantly higher than S11, especially during night times. This explains the higher $NO_3$ radical concentration in regions such as those near the west coast of the Bay of Bengal shown in Figure 4(c).

Figure 5: Same in Fig. 5(c).

The differences are also mainly due to the different chemical mechanisms. Carter's box-model simulation does directly report the difference of HCHO concentrations between S11 and S18. However, the $H_2O_2$ and $HO_2$ radical concentrations are significantly higher in S11 than those in S18, which suggests that HCHO, a major source of $H_2O_2$ and $HO_2$, is also higher in S11. This explains the lower HCHO in Figure 5(c).

Technical corrections

Line 152: No need to spell out SAPRC.
SAPRC was not spelled out in line 152 in the original manuscript.

Lines 169-171: This sentence is redundant in Lines 155-156.
The sentence is removed in the revised manuscript.

Line 252 (and some parts in this manuscript): "summer" should be "July".
Summer is replaced by July throughout the manuscript.

**Response to CC1:**
We appreciate the reviewer's feedback on the manuscript, and we carefully reviewed the comments and addressed each individually below, highlighting changes made in the revised manuscript.

General comments
In the context modeling demonstration of emission control effectiveness, the general understanding is that models and model inputs all have uncertainties. The work presented in the manuscript investigated two aspects of these: gas-phase chemical mechanisms and emissions inventories. To certain extent, other factors such as meteorological driving fields could cause even larger uncertainties.

We agree with the comment that uncertainties of other factors such as meteorological inputs can also lead to large differences in the model predictions. However, our WRF model performance for key surface parameters has been evaluated against observation data from NCDC, demonstrating good performance (Kang et al., 2021b). It is beyond the scope of the study to consider meteorological inputs from multiple regional weather forecast models.

To address these uncertainties, some well-established approaches, such as those used in the United States for regulatory air quality modeling purposes, do not use model results in an absolute sense as done in this work. Rather, the emissions control effectiveness is assessed using the combination of observations and modeled relative changes. The appropriateness of the methodology in this work seems to be questionable.

We agree with the comment regarding how the models are used for regulatory applications, such as attainment demonstrations, in the United States. In fact, as discussed in section 2.4, for each emission/inventory combination, we conducted 5 NOx reduction simulations (20, 40, 50, 60, and 80% reduction), 5 VOC reduction simulations, and 5 combined NOx and VOC reduction simulations. Overall, a total of 80 simulations (15 simulations * 5 emission/inventory combinations + 5 base cases) were conducted to assess how the predictions response to relative emission changes.

Specific comments

Section 2.1: It will be good to add a table summarizing main features for CS07, S11, S18 mechanisms, for example, the number of species, the number of reactions, major updates.

Thanks for the suggestion. Table S1 in the Supplementary Materials summarizes the main features of the three mechanisms.

Line 159: CMAQ also includes SAPRC07 mechanism, is CS07 different from the SAPRC07 mechanism in the CMAQ model?

The CS07 is different from the standard SARPC07 mechanism in the CMAQ model. The CS07 is a condensed version of SAPRC-07 comparable in size to CB05. It incorporates the condensed, approximate peroxy radical lumped operator method used in SAPRC99, CB4, and CB05. CS07 provides predictions of ozone ($O_3$), total peroxy radicals (PANs), and hydroxyl (OH) radicals that closely resemble those of the uncondensed mechanism (Carter, 2010).

Line 165: Is there any reason why S18-MEIC is not simulated?

The MEIC only has five emission sectors, making it difficult to re-speciate the emissions for the S18 mechanism. Specifically, solvent utilization emissions are not represented in the public version of MEIC as a separate sector (Wang et al., 2018). Since solvent utilization accounts for a significant fraction of VOC emissions in urban areas and has very different emission characteristics than fuel combustion sources, re-speciating the five-sector MEIC VOC emissions will be inaccurate.

The above discussion is included in the revised manuscript on page 7, lines 180-186.

Section 2.4: Are emission reductions limited to anthropogenic emissions, or it also applies to biogenic and fire emissions? The last sentence in this section needs some clarifications.

Thanks for the question. The emission reductions are applied to the final combined emissions, which include anthropogenic, biogenic and fire emissions.

Page 7, first paragraph: Since inorganic aerosols are also investigated, in the emissions comparisons, particulate matter emissions should be included as well.

Thanks for the comment. The inorganic aerosols in the model are dominantly from secondary formation. The impact of differences in the primary particle emissions on secondary inorganic aerosol predictions is very small. Thus, we did not include the primary particle emissions in the comparison.

Section 3.1, model performance evaluation: is there any PM2.5 chemical component measurement that can be used to evaluate model performance for inorganic aerosols (i.e., ammonium sulfate and ammonium nitrate). Also how does the model perform for NOx, SO2, or selected VOCs (if possible)?

Unfortunately, we do not have enough data to assess the model performance of $PM_{2.5}$ component concentrations. Since the current study focuses on how the relative changes on the predicts due to emission changes instead of determining the accuracy of the emission inventories or chemical mechanisms, we did not perform model performance assessments for other species.

Line 242: The reference to the US EPA model performance criteria needs to be listed.

The reference to the US EPA performance criteria is revised to reference a more recent paper by Emery et al. (2017).

Figure 1: Why is O3-8hr noticeably lower over the yellow sea when comparing S11-REAS to S11-MEIC?

Ozone formation sensitivity regime over the yellow sea is likely NOx-limited as the VOC-sensitive urban plume is advected to the marine environment (Vermeuel et al., 2019). Furthermore, satellite observed HCHO/NOx column over the Yellow Sea is greater than 6, clearing indicating a NOx-limited regime (Li et al., 2021). Since the NOx emissions in the upwind regions in the REAS inventory are significantly lower than these in the MEIC inventory, this leads to reduced $O_3$ formation in the S11-REAS results compared to the S11-MEIC results.

Line 274-286 and Figure 2: Model difference is presented for SIA (secondary inorganic aerosol). It is probably clearer if ammonium sulfate and ammonium nitrate are presented separately instead of being in a combined SIA. It will also help explain the impact of emission reductions on SIA concentrations.

The ISORROPIA aerosol thermodynamics model in the CMAQ model outputs only nitrate, sulfate, and ammonium ion concentrations. Depending on the pH of the aerosol aqueous solution, sulfate can be in the form of $SO_4^{2-}$ or $HSO4^-$.  Since these components are mostly dissolved and do not necessarily exist in solid form, we chose not to further break down SIA into ammonium sulfate and ammonium nitrate.

In addition, SIA is not clearly defined in the manuscript. What is included in SIA? How are the model primary inorganic aerosols separated from SIA?

SIA is defined in the original manuscript in the caption of Figure 2. In the revised manuscript, SIA is spelled out on page 10, line 298 as "secondary inorganic aerosol".

Line 314, section 3.4: Is there any column concentrations data for HCHO that can be used to evaluate model performance for HCHO?

As the reviewer pointed out, the emissions control effectiveness is assessed using modeled relative changes, so we didn't compare the HCHO column concentrations with observations. Our predicted monthly average surface HCHO concentrations from S11-MEIC are similar to the surface observations made in China (Zhang et al., 2021).  This is added to the revised manuscript on page 12, lines 355-356.

Figure 6: Why does Shenzhen seem to be least responsive to emission reductions?

Ozone concentration in Shenzhen is lower than other cities included in Figure 6 and most O3 there is background O3 on typical days (Kang et al., 2023). Thus, it is less responsive to emission reductions than other cities.

This discussion is included in the revised manuscript on page 13, lines 399-402.

Line 385-386: The explanation of the impact of VOC control on SIA concentrations seems to be too brief. More in-depth discussion will be helpful. For example, sulfate and nitrate probably need to be separated; maybe even PAN formation needs to be in the discussion.

Indeed, both sulfate and nitrate formation can be affected by the changes in the VOC emissions and sulfate and nitrate can be studied separately. However, from NOx/VOC emission control perspective, both sulfate and nitrate concentrations are changed, leading to changes in the PM2.5 concentrations. It is not particularly useful to separate the two components. We agree with the reviewer that a more detailed study is needed to focus on the underlying mechanisms that lead to sulfate/nitrate increases when VOC emissions are reduced.

Line 460: Why NO3 radical concentrations are much higher in northern China than southern China?

NO3 radical is formed from the reaction of NO2 + O3. In northern China, O3 and NO2 concentrations are significantly higher, leading to high production rate of NO3. In addition, NO3 is lost due to photolysis reaction and reactions with NO2. Due to higher PM concentrations northern China, the loss of NO2 due to photolysis is slower. The reaction of NO3 with NO2 is temperature dependent. Lower temperatures in northern China allow a longer lifetime of NO3. In summary, the faster formation rate and the slower loss rate of NO3 in northern China led to higher NO3 concentrations.

Section 3.5.4. HCHO concentration decreases duo to NOx emissions reductions are attributed to decrease of OH and NO3 levels. A conclusion is drawn indicating that secondary formation is the dominant source of HCHO. But the reduction of VOC emissions lead to steeper decreasing of HCHO. The VOC emissions reductions also cause OH and NO3 level increases as stated in the previous sections. These seem contradictory to the dominant secondary source conclusion.

We appreciate the comment. However, there is no contradiction. First of all, if the HCHO is mostly primary, NOx reduction would not lead to significant decrease of HCHO as indicated in Figure 10. While Figure 10 shows reduction of HCHO with reduction of VOCs, this reduction is likely due to reduction of precursor VOCs, as HCHO is the oxidation product of most of the VOCs. Our results are also consistent with many previous studies such as those in Houston (ref). Furthermore, as shown in Figure S15, the highest concentrations of HCHO occur in rural areas in Southern China with high biogenic emissions and low anthropogenic emissions, which confirms the secondary nature of HCHO. These secondary HCHO can also be transport to downwind urban areas.

The above discussion is included in the revised manuscript.

**References**

Carter, W. P. L.: Development of a condensed SAPRC-07 chemical mechansim, 2010.

Carter, W. P. L.: Documentation of the SAPRC-18 mechansim, Report to the California Air Resources Board, 2020.

Carter, W. P. L., Jiang, J., Orlando, J. J., and Barsanti, K. C.: Derivation of atmospheric reaction mechanisms for volatile organic compounds by the SAPRC mechanism generation system (MechGen), Atmospheric Chem. Phys., 25, 199–242, https://doi.org/10.5194/acp-25-199-2025, 2025.

Emery, C., Liu, Z., Russell, A. G., Odman, M. T., Yarwood, G., and Kumar, N.: Recommendations on statistics and benchmarks to assess photochemical model performance, J. Air Waste Manag. Assoc., 67, 582–598, https://doi.org/10.1080/10962247.2016.1265027, 2017.

Hu, J., Li, X., Huang, L., Ying, Q., Zhang, Q., Zhao, B., Wang, S., and Zhang, H.: Ensemble prediction of air quality using the WRF/CMAQ model system for health effect studies in China, Atmospheric Chem. Phys., 17, 13103–13118, 2017.

Kang, M., Zhang, J., Zhang, H., and Ying, Q.: On the Relevancy of Observed Ozone Increase during COVID-19 Lockdown to Summertime Ozone and PM2.5 Control Policies in China, Environ. Sci. Technol. Lett., https://doi.org/10.1021/acs.estlett.1c00036, 2021a.

Kang, M., Zhang, J., Zhang, H., and Ying, Q.: On the Relevancy of Observed Ozone Increase during COVID-19 Lockdown to Summertime Ozone and PM2.5 Control Policies in China, Environ. Sci. Technol. Lett., 8, 289–294, https://doi.org/10.1021/acs.estlett.1c00036, 2021b.

Kang, M., Zhang, J., Cheng, Z., Guo, S., Su, F., Hu, J., Zhang, H., and Ying, Q.: Assessment of Sectoral NOx Emission Reductions During COVID-19 Lockdown Using Combined Satellite and Surface Observations and Source-Oriented Model Simulations, Geophys. Res. Lett., 49, e2021GL095339, https://doi.org/10.1029/2021GL095339, 2022a.

Kang, M., Hu, J., Zhang, H., and Ying, Q.: Evaluation of a highly condensed SAPRC chemical mechanism and two emission inventories for ozone source apportionment and emission control strategy assessments in China, Sci. Total Environ., 813, 151922, https://doi.org/10.1016/j.scitotenv.2021.151922, 2022b.

Kang, M., Zhang, H., and Ying, Q.: Enhanced summertime background ozone by anthropogenic emissions – Implications on ozone control policy and health risk assessment, Atmos. Environ., 120116, https://doi.org/10.1016/j.atmosenv.2023.120116, 2023.

Li, R., Xu, M., Li, M., Chen, Z., Zhao, N., Gao, B., and Yao, Q.: Identifying the spatiotemporal variations in ozone formation regimes across China from 2005 to 2019 based on polynomial simulation and causality analysis, Atmospheric Chem. Phys., 21, 15631–15646, https://doi.org/10.5194/acp-21-15631-2021, 2021.

Vermeuel, M. P., Novak, G. A., Alwe, H. D., Hughes, D. D., Kaleel, R., Dickens, A. F., Kenski, D., Czarnetzki, A. C., Stone, E. A., Stanier, C. O., Pierce, R. B., Millet, D. B., and Bertram, T. H.: Sensitivity of Ozone Production to NOx and VOC Along the Lake Michigan Coastline, J. Geophys. Res. Atmospheres, 124, 10989–11006, https://doi.org/10.1029/2019JD030842, 2019.

Wang, P., Ying, Q., Zhang, H., Hu, J., Lin, Y., and Mao, H.: Source apportionment of secondary organic aerosol in China using a regional source-oriented chemical transport model and two emission inventories, Environ. Pollut., 237, 756–766, https://doi.org/10.1016/j.envpol.2017.10.122, 2018.

Zhang, K., Duan, Y., Huo, J., Huang, L., Wang, Y., Fu, Q., Wang, Y., and Li, L.: Formation mechanism of HCHO pollution in the suburban Yangtze River Delta region, China: A box model study and policy implementations, Atmos. Environ., 267, 118755, https://doi.org/10.1016/j.atmosenv.2021.118755, 2021.

---

## Author Response (AR2)

**Response to Reviewer Comments**

**"Explanation of the SAPRC mechanism: In Lines 127-139, the explanations for SAPRC are fully provided, allowing us to understand the content. However, without these introductions, we can still refer to Lines 79-84 and Lines 107-111. I would like to kindly request a brief introductory section for wider readers.**

Response: The text on lines 79-84 is a summary of previous research, which used the SAPRC mechanism. The description of SAPRC mechanisms on lines 127-139 provides some key aspects of the SAPRC mechanism that are more suitable to be included in the methods section for readers who are unfamiliar with the SARPC mechanism. We feel this information is better included in the methods section than in the introduction section. Thus, no changes were made regarding this comment.

**Additionally, I noticed that the Introduction mainly focuses on the emission inventory and its mechanism specific to this study. Could you clarify the motivation and purpose of this study? I expected that other aspects, such as meteorological conditions and deposition, would also contribute to uncertainty in the modeling. For example, have there been any previous studies that have shown that the uncertainty in emissions and related mechanisms is a key factor? It is recommended to revise the Introduction accordingly.**

Response: The impact of meteorology uncertainties on air quality model predictions of ozone and PM2.5 has been studied by Gilliam et al. et al. (2015) and Wen et al. (2025), respectively. In the revised manuscript, we added citations to Gilliam et al. et al. (2015) and Wen et al. (2025), on lines 53-55.

In the original manuscript, we have already clearly pointed out that the motivation of this study is to understand the combined effects of uncertainties in chemical mechanism and emission inputs to predicted air quality, which has not been quantified before (page 5, lines 115-118). In the revised manuscript, this motivation is pointed out more clearly.

**Please carefully review the tracked version (ATC1.pdf) during this revision process. The incomplete highlights in this review caused delays in seeing responses and corresponding changes in the main text."**

Response: Thanks for pointing this out. We highlighted the changes in the manuscript clearly in this round of revision.